# Observable variations in human sex ratio at birth

Yanan Long[1,2,3], Qi Chen[4], Henrik Larsson[4,5], Andrey Rzhetsky[2,3,6]*

**1** Department of Chemistry, The University of Chicago, Chicago, Illinois, United States of America, **2** Department of Medicine, The University of Chicago, Chicago, Illinois, United States of America, **3** Institute of Genomics and Systems Biology, The University of Chicago, Chicago, Illinois, United States of America, **4** Department of Medical Epidemiology and Biostatistics, Karolinska Institutet, Stockholm, Sweden, **5** School of Medical Sciences, Örebro University, Örebro, Sweden, **6** Department of Human Genetics and Committee on Quantitative Methods in Social, Behavioral, and Health Sciences, The University of Chicago, Chicago, Illinois, United States of America

* andrey.rzhetsky@uchicago.edu

**Data Availability Statement:** Aggregate data associated with described analyses and R scripts used in this analysis are available at https://github.com/yananlong/SRB. In particular, https://git.io/Jg5zx for the list of counties with neighbours, https://git.io/Jg5iy for the number of male and

## Abstract

The human sex ratio at birth (SRB), defined as the ratio between the number of newborn boys to the total number of newborns, is typically slightly greater than 1/2 (more boys than girls) and tends to vary across different geographical regions and time periods. In this large-scale study, we sought to validate previously-reported associations and test new hypotheses using statistical analysis of two very large datasets incorporating electronic medical records (EMRs). One of the datasets represents over half ($\sim$ 150 million) of the US population for over 8 years (IBM Watson Health MarketScan insurance claims) while another covers the entire Swedish population ($\sim$ 9 million) for over 30 years (the Swedish National Patient Register). After testing more than 100 hypotheses, we showed that neither dataset supported models in which the SRB changed seasonally or in response to variations in ambient temperature. However, increased levels of a diverse array of air and water pollutants, were associated with lower SRBs, including increased levels of industrial and agricultural activity, which served as proxies for water pollution. Moreover, some exogenous factors generally considered to be environmental toxins turned out to induce higher SRBs. Finally, we identified new factors with signals for either higher or lower SRBs. In all cases, the effect sizes were modest but highly statistically significant owing to the large sizes of the two datasets. We suggest that while it was unlikely that the associations have arisen from sex-specific selection mechanisms, they are still useful for the purpose of public health surveillance if they can be corroborated by empirical evidences.

## Author summary

The human sex ratio at birth (SRB), usually slightly greater than 1/2, have been reported to vary in response to a wide array of exogenous factors. In the literature, many such factors have been posited to be associated with higher or lower SRBs, but the studies conducted so far have focused on no more than a few factors at a time and used far smaller

female newborns plus the exogenous factors derived from the EQI dataset for each county, and https://git.io/Jg5MJ for the county, sex and birthday for each mother–newborn pair.

**Funding:** A.R. was funded by the DARPA Big Mechanism program under ARO contract W911NF1410333, by National Institutes of Health grants R01HL122712, 1P50MH094267, and U01HL108634-01, and by a gift from Liz and Kent Dauten. Additional support for A.R. came from King Abdullah University of Science and Technology (KAUST), awards number FCS/1/4102-02-01, FCC/1/1976-26-01, REI/1/0018-01-01, and REI/1/4473-01-01. The funders had no role in study design, data collection and analysis, decision to publish, or preparation of the manuscript.

**Competing interests:** The authors have declared that no competing interests exist.

datasets, thus prone to generating spurious correlations. We performed a series of statistical tests on 2 large, country-wide health datasets representing the United States and Sweden to investigate associations between putative exogenous factors and the SRB, and were able to validate a set of previously-reported associations while also discovering new signals. We propose to interpret these results simply as public health indicators awaiting further empirical confirmation rather than as implicated in (adaptive) sexual selection mechanisms.

## Introduction

Because human male gametes bearing X or Y chromosomes are equally frequent (being produced by meiosis symmetrically partitioning two sex chromosomes), and because ova bear only X chromosomes, one would expect a sex ratio at conception of exactly $\frac{1}{2}$ [1]. Indeed, a recent study using fluorescent *in situ* hybridization and array comparative genomic hybridization showed that the sex ratio at conception (SRC) was statistically indistinguishable from $\frac{1}{2}$ [2]. Nevertheless, the apparent sex ratio at birth (SRB), also known as the secondary sex ratio, has been documented to significantly deviate from $\frac{1}{2}$ under various circumstances, suggesting that a proportion of embryos are lost between conception and birth.

At least three processes may affect the observed SRB. First, female-embryo pregnancies may terminate early in development, driving the SRB up. It has been documented that these excess female-embryo losses tend to occur primarily during the first and early-second trimesters of pregnancy. Second, male-embryo deaths would drive the apparent SRB down. Male-embryo losses have indeed been observed to occur during the late-second and third trimesters [3]. Third, SRB may be affected by peri-conceptual maternal hormonal levels [4, 5]. Past studies proposed that the SRB can fluctuate with time and may be driven by a number of environmental factors, such as chemical pollution, events exerting psychological stress on pregnant women (such as terrorist attacks and earthquakes), radiation, changes in weather, and even seasons of conception (Table 1).

While there are multiple studies which have observed the positive associations between air pollution and spontaneous abortion [15, 16], most of those conclusions based on analyses of relatively small samples (Table 1), severely curtailing their statistical power. In this study, we harnessed the power of 2 very large datasets: the MarketScan insurance claim data [17] in the United States (which records the health events of more than 150 million unique Americans, with more than 3 million unique newborns recorded between 2003 to 2011), and Sweden's birth registry data (covering the birth and health trajectories of over $\sim$ 3 million newborns from 1983 to 2013) [18]. Our present study is the first systematic investigation of numerous chemical pollutants and other environmental factors using large datasets from two continents.

## Methods

### Data

The IBM Health MarketScan dataset [17] represents 104, 565, 671 unique individuals and 3, 134, 062 unique live births. The Swedish National Patient Register [18] record health statistics for over ten million individuals, and 3, 260, 304 unique live births. We juxtaposed time-stamped birth events in the two countries with exogenous factor measurements retrieved from the US National Oceanic and Atmospheric Administration, the US Environmental Protection Agency (EPA), the Swedish Meteorological and Hydrological Institute and Statistics Sweden.

**Table 1. Exogenous factors reported in the literature to have an impact on the SRB [6, 14].** A "-" indicates that sample sizes were not mentioned in the articles reporting or reviewing the corresponding results.

| Exogenous Factor | Number of Studies | Sample Size |
|---|---|---|
| Dioxins [6] | 13 | 291 |
| Polychlorinated biphenyls (PCBs) [6] | 9 | 98 |
| 1,2-Dibromo-3-chloropropane (DBCP) [6] | 2 | 29 |
| Dichlorodiphenyltrichloroethane (DDT) [6] | 4 | 1623 |
| Hexachlorobenzene (HCB) [6] | 2 | 262 |
| Vinclozolin [6] | 1 | 95 |
| Multiple pesticides [6] | 5 | 382 |
| Lead [6] | 5 | 6566 |
| Methylmercury [6] | 1 | 4808 |
| Multiple metals [6] | 10 | 1015 |
| Non-ionizing radiation [6] | 12 | 2926 |
| Ionizing radiation [6] | 15 | 4959 |
| Seasonality [7, 8] | 2 | - |
| Ambient temperature [9–12] | 4 | - |
| Economic stress [13] | 1 | - |
| Terrorist attacks [14] | 2 | - |

We used a subset of the MarketScan data that contained information on livebirths between 2003 to 2011 with county information encoded in Federal Information Processing Standards (FIPS) codes and a family link profile indicating the composition of the households in the dataset. The date, geographic distribution, and the mothers of the newborns can be directly extracted from these datasets. For environmental factors, we used the Environmental Quality Index (EQI) data compiled by the United States Environmental Protection Agency [19, 20].

## Cluster analysis

In order to simplify subsequent analyses, we first performed hierarchical clustering analysis on the Spearman's rank correlation coefficients matrix ($\rho$), using the Ward's method, which reduced the the EQI dataset's dimensionality. We then used the R-language [21] package `pvclust` [22] to minimize the total within-cluster variance [23]. The resulting dendrogram and list of factors can be found in the SI. Each cluster contains at least two factors and is represented by the mean of all the elements in the cluster.

## Regression analysis

We used multilevel Bayesian logistic regression with random effects implemented in the R-language package `rstan` [24]. To facilitate model building, we used the R-language package `brms` [25] with default priors. Sampling was performed with the No-U-Turn sampler (NUTS) [26] with 500 warm-up steps and 1500 iteration steps with 28 Markov chains, of which the convergence was assessed using the $\hat{R}$ statistic [27]. The model for the $j^{\text{th}}$ factor (predictor) is given as follows:

$$\text{logit}\left(p_j\right) = \log\left(\frac{p_j}{1 - p_j}\right) = \alpha_{[k]j} + \boldsymbol{\beta}_j^{\text{T}} \mathbf{x}_j, \tag{1}$$

where $p_j$ is the probability that a newborn is male, $\mathbf{x}_j$ is the vector representing the $j^{\text{th}}$ factor, $\boldsymbol{\beta}_j$ their coefficients, and $\alpha_{[k]j}$ the intercept for the $k^{\text{th}}$ group-level, representing states or counties

in the US, and *kommuner* (municipalities) or *län* (counties) in Sweden, whenever applicable. The group-level effect was modeled for a single random effect by

$$\alpha_j = \mu_a + \eta_j, \tag{2}$$

$$\eta_j \sim \mathcal{N}(0, \sigma_\eta^2) \tag{3}$$

and for two random effects, representing e.g. state- and county-specific effects, by

$$\alpha_j = \mu_a + \eta_j + v_j, \tag{4}$$

$$\eta_j \sim \mathcal{N}(0, \sigma_\eta^2), \tag{5}$$

$$v_j \sim \mathcal{N}(0, \sigma_v^2), \tag{6}$$

where $\eta_j$ and $v_j$ are independent of each other and for all $j$ [28]. Moreover, we partitioned the independent variables into septiles, so that $\boldsymbol{\beta}_j \in \mathbb{R}^6$, with one regression coefficient for each of the six septiles other than the first, which was treated as baseline [29].

We applied logistic regression in two ways. First, to assess the effect of environmental factors, we regressed each of the individual factors' septiles against the SRB, with each sample point representing a county. Therefore, each septile, aside from the baseline, has a coefficient. Second, to test whether maternal diagnostic history (DX) affected the SRB, we regressed a DX's indicator variables against the SRB, with each sample point representing a newborn/ mother pair. For model selection in both cases, we performed repeated (10 times) 10-fold cross-validation and calculated the information criterion relative to the null model (where $\mathbf{x}_j = \mathbf{0}$, i.e. the model was comprised solely of the intercept). We computed the average difference in information criterion ($\Delta$IC) and standard error (SE) for each factor obtained from leave-one-out (LOO) cross-validation [30], and used the Benjamini–Yekutieli method to adjust for multiple comparisons [31].

## Univariate time-series analysis

To assess the effect of one-off, stressful events on the SRB, we used two different time series techniques. First, we fitted seasonal univariate autoregressive integrated moving average (sAR-IMA) models using the Box-Jenkins method [32], in conjunction with monthly (28-day periods) and weekly live birth data up to the event and then performed an out-of-sample prediction. An sARIMA model is given by

$$
\underbrace{\left(1 - \sum_{j=1}^{P} \Phi_j L^j\right)}_{\text{sAR}} \underbrace{\left[(1 - L^S)^D\right]}_{\text{sI}} \underbrace{\left(1 - \sum_{j=1}^{p} \phi_j L^j\right)}_{\text{AR}} \underbrace{\left[(1 - L)^d\right]}_{\text{I}} y_t
$$
$$
= \underbrace{\left(1 + \sum_{j'=1}^{Q} \Theta_{j'} L^{j'}\right)}_{\text{sMA}} \underbrace{\left(1 + \sum_{j'=1}^{q} \theta_{j'} L^{j'}\right)}_{\text{MA}} \varepsilon_t, \tag{7}
$$

where AR indicates the autoregression term, I the integration term, and MA the moving average term (an "s" before any of the above stands for "seasonal"). Moreover, $y_t$ indicates the observed univariate time series of interest, $L$ is the lag operator such that $L(y_t) = y_{t-1}$, $\varepsilon$'s white

noises, $S \geqslant 2$ the degree of seasonality (i.e., the number of seasonal terms per year, chosen to be 4 in our study), and $\phi$'s, $\theta$'s, $\Phi$'s, and $\Theta$'s are model parameters to be estimated. We used the `auto.arima` function from the R-language package `forecast` [33, 34] to fit the data, which performed a step-wise search on the $(p, d, q, P, D, Q)$ hyperparameter space and compared different models by using the Bayesian Information Criterion (BIC) [35]. We confirmed the optimalx models' goodness-of-fit using the Breusch-Godfrey test on the residuals, which tested for the presence of autocorrelation up to degree $S$ [36–38].

On the other hand, we fitted the same data as above to Bayesian structural time series (BSTS) models, which are state-space models given in the general form by [39]:

$$y_t = Z_t^\mathsf{T}\alpha_t + \varepsilon_t, \qquad \varepsilon_t \sim \mathcal{N}(0, H_t), \tag{8}$$

$$\alpha_{t+1} = T_t\alpha_t + R_t\eta_t, \qquad \eta_t \sim \mathcal{N}(0, Q_t), \tag{9}$$

where $y_t$ is the observed time series and $\alpha_t$ the *unobserved* latent state. In particular, we used the local linear trend model with additional seasonal terms [39, 40]:

$$y_t = \mu_t + \tau_t + \varepsilon_t \tag{10}$$

$$\mu_t = \mu_t + \delta_{t-1} + u_t \tag{11}$$

$$\delta_t = \delta_{t-1} + v_t \tag{12}$$

$$\tau_t = -\sum_{s=1}^{S-1} \tau_{t-s} + w_t. \tag{13}$$

Here, we define $\eta_t = \begin{bmatrix} u_t & v_t & w_t \end{bmatrix}$; $Q_t$ is a $t$-invariant block diagonal matrix with diagonal elements $\sigma_u^2, \sigma_v^2$ and $\sigma_w^2$. Finally, we denote $\alpha_t = \begin{bmatrix} \mu_t & \delta_t & \tau_{t-S+2} & \cdots & \tau_t \end{bmatrix}$, which implies that both $Z_t$ and $T_t$ are $t$-invariant matrices of 0's and 1's such that Eqs 10–13 hold. We used the R package `CausalImpact` [41], which in turn relied on the R package `bsts` [42] as backend, to fit the data.

## Correlation and causality

To test whether the SRB was effected by ambient temperature, we grouped daily SRB data and temperatures into 91-day (13-week) periods and calculated the Pearson correlation coefficient ($r$) between each SRB and ambient temperature. We then performed the Student's $t$-test for the null hypothesis that the true correlation is 0. Furthermore, we fitted the SRB/temperature pair to a vector autoregression (VAR) model for a maximum lag order of 4 (52 weeks), using the BIC as the metric for model selection, and then tested for the null hypothesis of the non-existence of Granger causality using the $F$-test [43].

## Results

We start by describing the negative results (i.e. a lack of a significant association), concordant across the two datasets. Our model selection rejected the whole spectrum of models that allow for periodic, annual SRB changes [7, 8]. For both US and Swedish datasets, the best-fitting model described the SRB as lacking seasonality throughout the year. Similarly, when we tested the claim that ambient temperatures during conception affect the SRB [9–12], we found that neither dataset supported this association. Both the Student's $t$-test and the $F$-test concluded

that the SRB was independent of ambient temperature measurements (Table G in S1 Appendix).

A comparison of each dataset's environmental measurements revealed that Sweden enjoyed both lower variations and lower mean values of measured concentrations of substances in the air. Unfortunately, the Swedish dataset also provided fewer measured pollutants, which made our cross-country analysis more difficult. Fig 1A shows a comparison of pollutant concentration distributions in both countries. The US environmental measurements dataset presented its own difficulty, as many pollutants appeared highly collinear in their spatial variation. To address this, we performed a cluster analysis on the environmental factors, subdividing them into 26 clusters (Table 2 and Fig B in S1 Appendix). All pollutants within the same cluster were highly correlated, while the correlation between distinct clusters was much smaller, allowing for useful association inferences between SRB changes and environmental states.

Using the US dataset, we were able to validate the findings of a number of previous studies regarding the association between the SRB and exogenous factors (Table 3). Specifically, our data suggests that aluminium (Al) in air, chromium (Cr) in water and total mercury (Mg) quantity drive the SRB up, while lead (Pb) in soil appears to be associated with a decreased SRB. Meanwhile, we have found no evidence for a number of previous reports, indicated with a dash in the second column in Table 3. We also established several new environmental associations that have not been reported before (Table 4, Figs 1B–1M and 2). Fig 1B–1M show that increased pollutant levels appear to be associated with both increased and decreased SRB values (Plates **E,F,H,I**, and **J**, and the remaining Plates, respectively). In the case of PCBs (polychlorinated biphenyls), on which the literature has reported conflicting evidences [44], we found a positive correlation with the SRB. Since the sample sizes of the studies published thus far were very small (cf. Table 1), our PCBs result would have substantially larger statistical power.

The geographic distribution of these pollutants varies remarkably, as seen in Fig 2. For example, lead in land (Fig 2**B**) appears to be enriched in the northeast, southwest, and mideast US, but not in the south. Hydrazine (Fig 2**F**) appears to follow capricious, blotch-like shapes in the eastern US, each blotch likely centered at a factory emitting this pollutant. Total mercury deposition in water (Fig 2**E**) mostly affects eastern US states with the heaviest load in the northeastern states. It is this variability in the environmental distribution of various substances that allowed us to tease out these individual associations.

Finally, when we tested links between two stressful events in the US (Hurricane Katrina and the Virginia Tech shooting) and the SRB using seasonal autoregressive integrated moving-average (sARIMA) models and state-space models (SSMs) (see the *Univariate time-series analysis* section in *Methods*), we were able to identify significant associations only in the case of the Virginia Tech shooting—the SRB was lower than expected 34 weeks after the event (see Figs 3**C** and 4**C**, cf. Tables E(c) and F(c) in S1 Appendix).

## Discussion

While SRB fluctuations in space and time are well-documented and non-controversial, there is a diverse range of competing theories striving to explain SRB changes in terms of mechanistic selective pressure [45]. The most frequently mentioned theory is the Trivers–Willard hypothesis (TWH), named after the researchers who proposed it [46]). The TWH postulates that, because the cost of rearing children is much higher for females than for males, in favourable, resource-rich environments, males would have more offspring than females, and vice versa in unfavourable conditions. Natural selection would then favour individuals with higher fitness, where fitness is equated to individuals' reproductive success (in this case, the number of

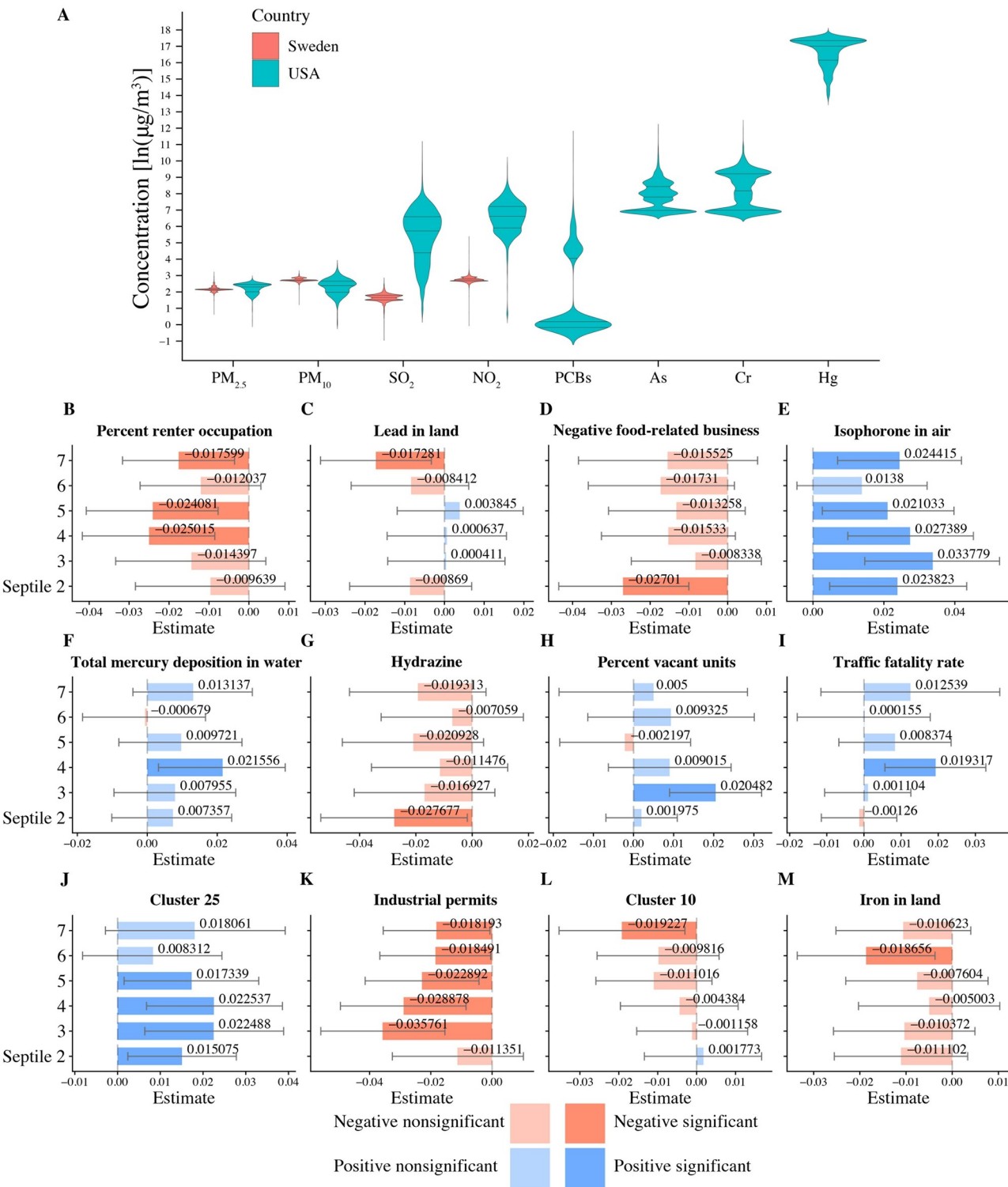

**Fig 1. Airborne health-related substances and their association with the SRB. A**: Comparison of airborne pollutant concentrations across the US (cyan violin plots) and Sweden (pink violin plots). Only 4 air components, fine particulate matter ($PM_{2.5}$), coarse particulate matter ($PM_{10}$), sulfur dioxide ($SO_2$), and nitrogen dioxide ($NO_2$) are measured in both countries. US counties appear to have higher mean pollution levels and are more variable in terms of pollution. **B-M**: A sample of 12 one-environmental factor logistic regression models that are most explanatory with respect to SRB. For each environmental factor, we partition counties into 7 equal-sized groups (septiles), ordered by levels of measurements, so that the first septile corresponds to the lowest and the highestnth septile to the highest concentration. Each plot shows bar plots of regression coefficients and 95% confidence intervals (error bar) of the second to the seventh septiles, with the first septile chosen as the reference level. We rank the 12 models by the

statistically significant factor's association strength with at least one statistically significant coefficient by decreasing ΔIC; septiles whose coefficients are not significantly different from 0 at the 95% confidence level have been plotted with a reduced alpha level. Blue bars represent positive coefficients, whereas red bars represent negative coefficients. "Negative food-related businesses" is a term used by the Environmental Protection Agency's Environmental Quality Index team and is explained as "businesses like fast-food restaurants, convenience stores, and pretzel trucks." "Percent vacant units" stands for "percent of vacant housing units." Substances contributing to clusters 10 and 25 are listed in Table 2. See Table K in S1 Appendix for more details regarding the factors' and clusters' identities.

offspring reaching reproductive age). According to the TWH, natural selection pushes the SRB up (more males) in favourable conditions, and down (more females) in unfavourable environment.

More explicitly stated, the TWH depends on the following three assumptions [5, 46, 47]:

**Assumption A1**. The condition of a mother during parental investment is correlated with the condition of her offspring; in other words, mothers in better conditions have offspring that will be in better conditions.

**Assumption A2**. The condition of the offspring persists after parental investment ends, and is positively correlated with the offspring's reproductive success.

**Assumption A3**. Males have larger variability in reproductive success than females and, as a result, they are more susceptible to sexual selection.

From these assumptions the TWH makes the following *deductive* inference on SRB variability:

**Conclusion C1**. The SRB varies such that females in favourable conditions have more male offspring, and in unfavourable conditions, more female offspring.

Assumption A3 is called Bateman's principle [48] (BP), and was suggested in a classic fruit fly genetics study on sexual selection. The original experimental results with *Drosophila mela-nogaster* indicated that males benefited more from multiple mating than females in terms of fitness. This asymmetry was thought to have originated in anisogamy, which means that a sperm is much smaller than an ovum and therefore requires less resources. Unfortunately, this result was never replicated (see [49, 50] for critiques of Bateman's methodology). Nevertheless, a modified version of BP, which generalizes anisogamy to parental investment, has enjoyed prominence among evolutionary biologists [51]. One of the critiques of BP claims that male cost of reproduction is in reality much higher than suggested by Bateman. This is because Bateman failed to account for the fact that males do not produce sperms stoichiometrically to match the number of female-produced ova. Instead, they produce semen, a mixture of a very large number of male gametes and accompanying secretions, rich in nutrients and other substances beneficial to reproduction [52]. Therefore, once the full range of investment patterns across life history (e.g. intrasex competition, secondary sexual characteristics, territorial defence) has been taken into account, it is unclear if reproductive investments of females exceed those of males [53, 54].

Faced with such criticisms as well as an increasing amount of evidence from species across the animal kingdom that did not conform to BP [55], supporters of BP have responded that sex differences ultimately originated from *historical* anisogamy [56, 57], and that there have also been subsequent ecological factors independent of anisogamy that drove sexual dimorphism having to do with resource competition between the sexes, which may not result in stronger selection on males [58, 59]. Moreover, as a counter-challenge to the former point, supporters of BP also refer to aggregate results in favour of BP, including a phylogenic meta-analysis by Janicke et al. in which significant differences in reproductive success variances in species across the animal kingdom were found [60]. This reworking allowed for a potential remedy for BP, namely by generalizing it as follows [55, 61, 62]:

**Table 2. Pollutant clusters discovered by applying the Ward's method to the EQI raw measurements dataset.**

| Cluster number | factor |
|---|---|
| 1 | a_hcbd_ln,a_hccpd_ln |
| 2 | a_nitrobenzene_ln,a_dma_ln |
| 3 | a_2clacephen_ln,a_bromoform_ln |
| 4 | a_pnp_ln,a_toluene_ln |
| 5 | a_be_ln,a_se_ln |
| 6 | a_dmf_ln,a_edb_ln,a_edc_ln |
| 7 | a_teca_ln,a_procl2_ln,a_cl4c2_ln,a_vycl_ln,county_pop_2000 |
| 8 | a_benzyl_cl_ln,a_me2so4_ln |
| 9 | mean_zn_ln,mean_cu_ln |
| 10 | mean_al_pct,mean_p_pct |
| 11 | numdays_close_activity_tot,numdays_cont_activity_tot |
| 12 | mean_as_ln,mean_se_ln |
| 13 | a_glycol_ethers_ln,a_etn_ln,a_vyac_ln |
| 14 | mean_na__pct_ln,mean_mg_pct_ln,mean_ca_pct_ln |
| 15 | a_cs_ln,a_edcl2_ln |
| 16 | a_ccl4,a_mtbe_ln |
| 17 | pct_harvest_acres,herbicides_ln,insecticides_ln |
| 18 | a_112tca_ln,a_ch3cn_ln |
| 19 | a_hcb_ln,a_pcp_ln,a_pcbs_ln |
| 20 | mg_ln_ave,k_ln_ave |
| 21 | pct_defoliate_acres_ln,pct_disease_acres_ln,pct_nematode_acres_ln |
| 22 | a_so2_mean_ln,a_no2_mean_ln,a_o3_mean_ln,so4_mean_ave |
| 23 | med_hh_value,med_hh_inc |
| 24 | rate_food_env_pos_log,rate_rec_env_log |
| 25 | ca_ln_ave,nh4_mean_ave |
| 26 | w_as_ln,w_ba_ln,w_cd_ln,w_cr_ln,w_cn_ln |
| | w_fl_ln,w_hg_ln,w_no3_ln,w_no2_ln,w_se_ln |
| | w_sb_ln,w_be_ln,w_ti_ln,w_endrin_ln |
| | w_lindane_ln,w_methoxychlor_ln,w_toxaphene_ln |
| | w_dalapon_ln,w_deha_ln,w_oxamyl_ln,w_simazine_ln |
| | w_dehp_ln,w_picloram_ln,w_dinoseb_ln |
| | w_hccpd_ln,w_carbofuran_ln,w_atrazine_ln |
| | w_alachlor_ln,w_heptachlor_ln,w_heptachlor_epox_ln |
| | w_24d_ln,w_silvex_ln,w_hcb_ln,w_benzoap_ln |
| | w_pcp_ln,w_124tcib_ln,w_pcb_ln,w_dbcp_ln |
| | w_edb_ln,w_xylenes_ln,w_chlordane_ln,w_dcm_ln |
| | w_odcb_ln,w_pdcb_ln,w_vcm_ln,w_11dce_ln |
| | w_t12dce_ln,w_edc_ln,w_111trichlorane_ln |
| | w_ccl4_ln,w_pdc_ln,w_trichlorene_ln,w_112tca_ln |
| | w_c2cl4_ln,w_cl1benz_ln,w_benzene_ln,w_toluene_ln |
| | w_ethylbenz_ln,w_stryene_ln,w_alpha_ln,w_dce_ln |

**Assumption A3**\*. The sex with the larger reproductive success variance is more susceptible to sexual selection.

From this, the generalized version of the TWH follows:

**Table 3. Test results for factors selected from the literature reports (Table 1).** We included a factor only if both its ΔIC and the coefficient of at least one of its septiles was statistically significant.

| Factor name | effect |
|---|---|
| PCBs (air and water) | ↑ |
| DBCP (water) | − |
| Lead (land) | ↓ |
| Lead (air) | − |
| Aluminium (air) | ↑ |
| Chromium (air) | − |
| Chromium (water) | ↑ |
| Arsenic (land) | − |
| Arsenic (water) | ↑ |
| Cadmium (air and water) | − |
| Total mercury deposition | ↑ |
| Violent crime rate | − |
| Unemployed rate | − |
| Working out of county (long commute) | − |

**Conclusion C1**[*]. The SRB should vary such that females in favourable conditions have more offspring of the sex more susceptible to sexual selection, and in unfavourable conditions, more offspring of the sex less susceptible to sexual selection.

This version of BP is consistent with "sex-role reversals" observed in many species, in which females exhibit larger susceptibility to sexual selection. In addition, it allows for sufficient flexibility such that the identity of "the sex more susceptible to sexual selection" may be influenced by exogenous conditions [58]. Candidates for the identity of that sex include higher variance in number of (adult) offspring and higher variance in parental investments [5, 63].

**Table 4. Test results for additional factors with statistically significant effects.** We included a factor only if both its ΔIC and the coefficient of at least one of its septiles was statistically significant.

| Factor name | effect |
|---|---|
| Iron | ↓ |
| Nitrate | ↑ |
| 2-Nitropropane | ↑ |
| Carbon monoxide | ↑ |
| Bis-2-ethylhexyl phthalate | ↓ |
| Ethyl chloride | ↑ |
| Isophorone | ↑ |
| Hydrazine | ↓ |
| Phosphorus | ↑ |
| Quinonline | ↓ |
| Extreme drought | ↑ |
| Traffic fatality rate | ↑ |
| Industrial permits per 1000 km of stream | ↓ |
| Animal units | ↓ |
| Irrigation | ↓ |
| Negative food related businesses | ↓ |
| Renter occupation | ↓ |
| Vacant units | ↑ |

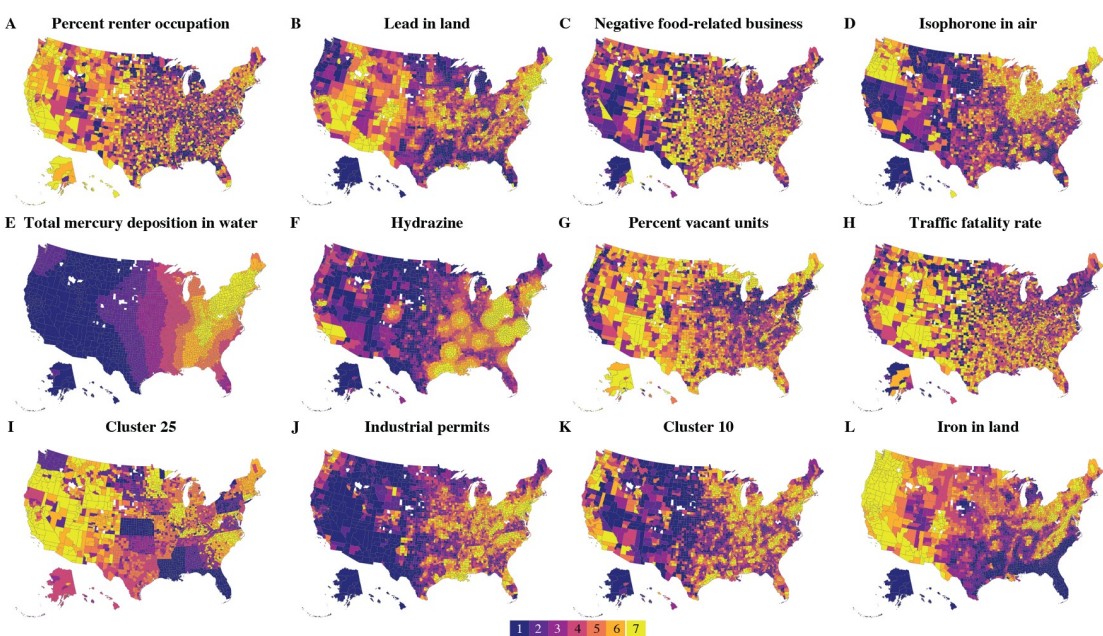

**Fig 2. County-level geographical septile distribution for the first 12 statistically significant factors with at least one statistically significant coefficient ranked by decreasing ΔIC.** The factors labelled **A–M** are the same as shown in Fig 1, Plates **B–M** and are ordered identically in both figures. Base map was taken from https://github.com/hrbrmstr/albersusa/blob/master/inst/extdata/composite_us_counties.geojson.gz.

Nevertheless, under this revised framework, for sexually dimorphic selection patterns to develop and persist as opposed to randomly fluctuating across time [64], one inevitably has to invoke the sexual cascade hypothesis: a small initial difference (e.g. anisogamy) in sex-related phenotype will "snowball" into larger, persistent patterns through hereditary feedback loops

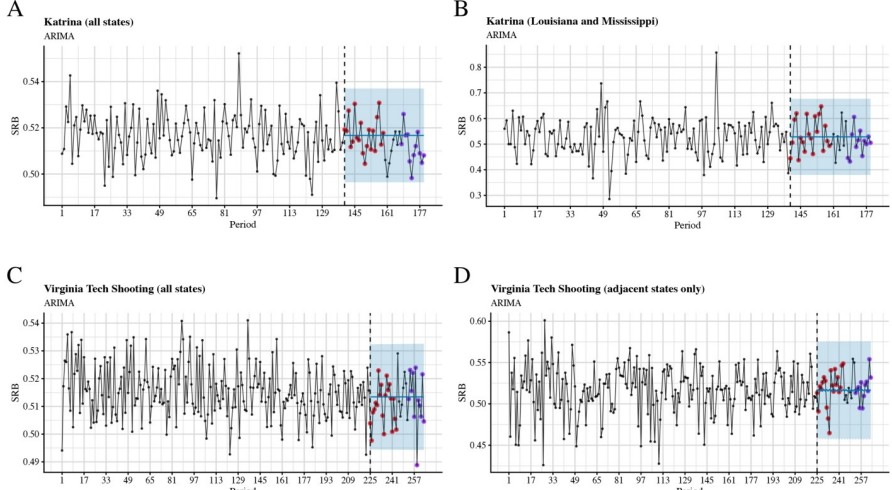

**Fig 3. Time series plots and out-of-sample forecasts for SRB data grouped into 7-day periods and fitted with seasonal ARIMA models.** The blue shade is the 95% confidence level. The observed SRBs for the first five months after the intervention are presented by red dots, whereas the observed SRBs for 7 to 9 months after the intervention are presented by purple dots. A: Hurricane Katrina, all states; B: Hurricane Katrina, Louisiana and Mississippi only; C: Virginia Tech shooting, all states; D: Virginia Tech shooting, adjacent states only.

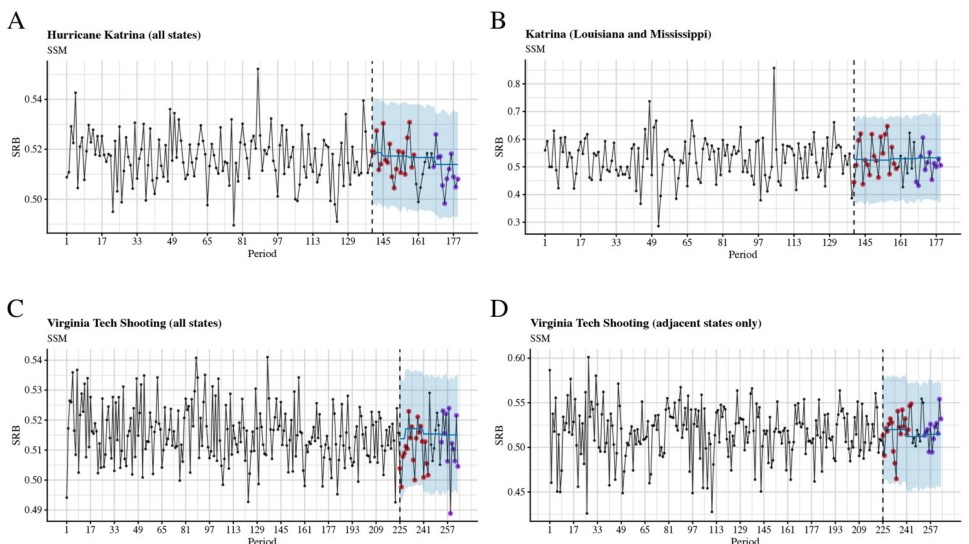

**Fig 4. Time series plots and out-of-sample forecasts for SRB data grouped into 7-day periods and fitted with state space models.** The blue shade is the 95% confidence level. The observed SRBs for the first five months after the intervention are presented by red dots, whereas the observed SRBs for 7 to 9 months after the intervention are presented by purple dots. A: Hurricane Katrina, all states; B: Hurricane Katrina, Louisiana and Mississippi only; C: Virginia Tech shooting, all states; D: Virginia Tech shooting, adjacent states only.

[65–67]. Such cascading has also featured in the above-mentioned meta-analysis discussion regarding high-level explanatory patterns among animal species [60], bracketing all differences in sex-related traits into one-dimensional sexual selection [55]. There is a plethora of other competing theories, e.g. [64] and [62], which predict largely stochastic variations of sex-related phenotypes, emphasizing the role social and ecological factors have played in shaping *plastic* sex-roles [55, 68, 69]. Even if the last point may still be somewhat contentious [66], BP and (by extension) the TWH are, at the very least, not the only game in town when it comes to explaining and predicting patterns related to sexual selection: male and female phenotypes of a given species in a given environment are most likely the results of a large number of exogenous factors without any single one of them being particularly dominant [70, p. 177].

One key ramification of the above analysis is that the TWH cannot provide a comprehensive account of the range of exogenous factors associated with SRB variation under the kind of circumstances present in our study. Further, the empirical success of the TWH is mixed, with only 50% of studies confirming it, and around 20% of studies producing statistically significant results in the opposite direction [47], which is consistent with our finding that many different pollutants might be assumed to be "bad" for mothers (e.g. pollutants, traffic fatality rates, junk food) had associations with SRB in *opposite* directions. The scepticism against the applicability of the TWH in contemporary human societies is further strengthened by two recent population studies in Sweden with large sample sizes (4.7 and 5.7 million live births, respectively), which found no SRB heritability [71, 72]. In particular, Zietsch et al. have demonstrated that there exists neither within-individual SRB auto-correlation (contra Assumption A1) nor similarity in the SRB for children of siblings (contra Assumption A2). They also concluded that within-family SRB was associated with the final family size, suggesting that SRB variations may have been the result of SRB-*aware* family planning [71]. Taken together, such evidence also places other adaptive (i.e. via heritable sexual selection) theories explaining SRB variations, such as adaptive versions of hormonal hypothesis [63], maternal dominance hypothesis [73,

74] and the Bruce effect [75] in the same predicament. Appealing to evolutionary history (i.e. TWH was in operation in the past but not at present, or TWH is an effect of some vestigial evolutionary mechanism) is of no help here, since an adaptive selection mechanism cannot explain why and how, at some point in history, the heritability was lost [76]. In other words, if SRB is ever influenced by some factor(s) at least partially heritable, then SRB itself would have to be heritable as well, which the results from Zietsch et al. rule out. Thus, our results are better interpreted as supporting the overwhelming influence of random Mendelian segregation on the SRB (cf. [77] which claims *complete* attribution of SRB variation to Mendelian segregation in some non-human species), such that SRB variations are at least primarily due to *non-adaptive* (e.g. socio-cultural [71, 78, Ch. 14]) causal factors, possibly including those common to both changes in the SRB and associated exogenous factors.

By way of conclusion, we note that the literature includes substantial reports on the relationship between the SRB and public health [3], and we would like to consider the question of whether the SRB can be used as an indicator for public health events and, if so, whether the relationship between the SRB and certain diseases reveals causal relationships. As the preceding discussion demonstrates, even if the existence of *adaptive* causal relationships between environmental factors and the SRB may be unlikely (contra [79]), *associations*—including the ones presented in this work—may be used as signals for (adverse) public health conditions, as long as they are established experimentally. To this end, we reiterate that there *are* agreements between the associations established in our work and those in the literature [6, 14], and that our results do support the non-monotonous, dose-response profiles frequently reported in the literature [44] (Table 3). Therefore, future research programmes might instead focus on exploring and validating the associations between SRB and environmental factors that reliably *predict* adverse public health effects for certain subpopulations [72] using large datasets with covariates sampled frequently across considerable spatio-temporal ranges [71]. Another interesting direction would be to determine the potentially non-adaptive physiological mechanisms.

## Limitations

Unlike some of the recent studies [80], we did not have access to the sex of stillbirths, which would have enabled us to probe negative selection *in utero* against frail males [3]. When quantifying pollutants in the US, we used the EPA air quality raw data, which was an average of measurements taken over a short period of time, rather than over years or decades, which would have enabled long-term and causal analyses. Neither did it include information for individual exposures to those factors, which might render a straightforward interpretation of our results subject to ecological fallacies. Finally, the subjects in our US study were commercially-insured and had medical claims, which likely came from a different probability distribution to the general population in the US.

## Supporting information

**S1 Appendix. Additional results, figures and tables.** Fig A. Distribution of the SRB in the US and Sweden at the county level (US) or the kommun level (Sweden). Fig B. Dendrogram with statistically significant clusters (95% level) in red boxes. Table A. Differences in information criterion (ΔIC) and their standard errors (SE) of individual factors with fixed-effect only. Non-significant factors are omitted. Table B. Differences in information criterion (ΔIC) and their standard errors (SE) of individual factors with fixed-effect only. Non-significant factors are omitted. Fig C. Time series plots and out-of-sample forecasts for SRB data grouped into 28-day periods and fitted with seasonal ARIMA models. The blue shade is the 95% confidence

level. The observed SRBs for the first 5 months after the intervention are presented by red dots, whereas the observed SRBs for 7–9 months after the intervention are presented by purple dots. See also Table C. Table C. Out-of-sample forecasts for the first 10 months after the intervention using SRB data grouped into 28-day periods and fitted with seasonal ARIMA models. Any period of which the observed SRB is outside of the 95% confidence level is marked by an asterisk (*). Figure D. Time series plots and out-of-sample forecasts for SRB data grouped into 28-day periods and fitted with state-space models. The blue shade is the 95% confidence level. The observed SRBs for the first 5 months after the intervention are presented by red dots, whereas the observed SRBs for 7–9 months after the intervention are presented by purple dots. See also Table D. Table D. Out-of-sample forecasts for the first 10 months after the intervention using SRB data grouped into 7-day periods and fitted with state-space models. Any period of which the observed SRB is outside of the 95% confidence level is marked by an asterisk (*). Fig E. Time series plots and out-of-sample forecasts for SRB data grouped into 28-day periods and fitted with seasonal ARIMA models. The blue shade is the 95% confidence level. The observed SRBs for the first 5 months after the intervention are presented by red dots, whereas the observed SRBs for 7–9 months after the intervention are presented by purple dots. See also Table E. Table E. Out-of-sample forecasts for the first 10 months after the intervention using SRB data grouped into 28-day periods and fitted with seasonal ARIMA models. Any period of which the observed SRB is outside of the 95% confidence level is marked by an asterisk (*). Fig F. Time series plots and out-of-sample forecasts for SRB data grouped into 28-day periods and fitted with state-space models. The blue shade is the 95% confidence level. The observed SRBs for the first 5 months after the intervention are presented by red dots, whereas the observed SRBs for 7–9 months after the intervention are presented by purple dots. See also Table F. Table F. Out-of-sample forecasts for the first 10 months after the intervention using SRB data grouped into 7-day periods and fitted with state-space models. Any period of which the observed SRB is outside of the 95% confidence level is marked by an asterisk (*). Table G. $p$-values for $t$- and $F$-tests on the correlation between Sweden's SRB and temperature and precipitation in Sweden. Table H. Differences in information criteria ($\Delta$IC) and their standard errors (SE) of individual factors at the kommun (municipality) level, with random effect at the län (county) level. Table I. Differences in information criteria ($\Delta$IC) and their standard errors (SE) of individual factors at the län (county) level. Table J. Contingency table of maternal diagnosis history versus the sex of livebirths. Table K. List of variable names used in the main text and their corresponding definitions and units (if applicable).
(PDF)

## Acknowledgments

We are grateful to E. Gannon and M. Rzhetsky for comments on earlier versions of this manuscript.

## Author Contributions

**Conceptualization:** Yanan Long, Henrik Larsson, Andrey Rzhetsky.

**Data curation:** Yanan Long, Qi Chen, Henrik Larsson.

**Formal analysis:** Yanan Long, Qi Chen, Andrey Rzhetsky.

**Funding acquisition:** Henrik Larsson, Andrey Rzhetsky.

**Investigation:** Yanan Long, Henrik Larsson, Andrey Rzhetsky.

**Methodology:** Andrey Rzhetsky.

**Project administration:** Andrey Rzhetsky.

**Resources:** Henrik Larsson, Andrey Rzhetsky.

**Software:** Yanan Long, Qi Chen.

**Supervision:** Andrey Rzhetsky.

**Validation:** Qi Chen.

**Visualization:** Yanan Long.

**Writing – original draft:** Yanan Long, Henrik Larsson, Andrey Rzhetsky.

**Writing – review & editing:** Yanan Long, Qi Chen, Henrik Larsson, Andrey Rzhetsky.

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
