## [Decision Letter · Decision Letter 0]

30 Aug 2021

Dear Dr Rzhetsky,

Thank you very much for submitting your manuscript "Observable Variations in Human Sex Ratio at Birth" for consideration at PLOS Computational Biology. As with all papers reviewed by the journal, your manuscript was reviewed by members of the editorial board and by several independent reviewers. The reviewers appreciated the attention to an important topic. Based on the reviews, we are likely to accept this manuscript for publication, providing that you modify the manuscript according to the review recommendations.

I apologise for the long wait for reviews, with which there were some difficulties.

Both reviewers recommended revisions, and I would like to invite you to submit a revised manuscript that has considered and responded to the reviewers’ points.

I do not agree with Reviewer 2’s comments about your ‘confusion’ as to evolutionary past vs. present, nor their claim the Zietsch et al. results do not bear on the T-W assumptions you laid out. Zietsch et al. (2021 https://doi.org/10.1098/rspb.2021.0304 ) provided a response to the cited commentary, and this article may help to clarify the issue. I think your treatment of T-W was very nice and clear, but you may wish to briefly address this issue in your revision so as other readers don’t have the same question.

Sincerely,

Brendan Zietsch

Guest Editor

PLOS Computational Biology

Nina Fefferman

Deputy Editor

PLOS Computational Biology

[LINK]

Dear Dr Rzhetsky,

Thank you for submitting your work to PLoS Computational Biology. I apologise for the long wait for reviews, with which there were some difficulties.

I have received reviews from two experts and have carefully read the paper myself. Both reviewers recommended revisions, and I would like to invite you to submit a revised manuscript that has considered and responded to the reviewers’ points.

I do not agree with Reviewer 2’s comments about your ‘confusion’ as to evolutionary past vs. present, nor their claim the Zietsch et al. results do not bear on the T-W assumptions you laid out. Zietsch et al. (2021 https://doi.org/10.1098/rspb.2021.0304 ) provided a response to the cited commentary, and this article may help to clarify the issue. I think your treatment of T-W was very nice and clear, but you may wish to briefly address this issue in your revision so as other readers don’t have the same question.

Kind regards,

Brendan Zietsch

Reviewer's Responses to Questions

**Comments to the Authors:**

Reviewer #1: The authors explored the factors that underlie deviations in the human sex ratio in humans from Sweden and the United States of America. The authors used a large dataset (> 150 million people) to discover that increased levels of a number of pollutants affect human sex ratios at birth. These pollutants could induce higher or lower sex ratios at birth, depending on the pollutant. It is a well written paper on an interesting topic and deserving of publication in PLOS Computational Biology. I only have some very minor comments.

1. The analyses are complicated. Why did you choose a bayesian approach? I’m not critical of the statistic approach taken. However, a justification for the statistical approach would be great to see in the methods.

2. There are quite a few parameters in your model. Could overparameterization be an issue here? Granted, the sample size is very large.

3. Could you please provide the R code used to analyse this data

4. Could you also assess mean income across the populations in your models or another socioeconomic measure? Or can you justify why that is unnecessary or irrelevant?

Reviewer #2: Review of “Observable Variations in Human Sex Ratio at Birth” by Yanan Long, Qi Chen, Henrik Larsson, and Andrey Rzhetsky

The authors present analyses of seasonal, social, and environmental influences on the sex ratio at birth. Their analyses are based upon two large databases (IBM and Sweden). Each contains about three million live births. They report that there is no influence of season on the sex ratio at birth. They also report significant associations between the sex ratio at birth and the level of various social “factors” (e.g., traffic fatality rate) and environmental factors (pollutants). The data on social factors comes from several sources, including US NOAA, EPA See Table S11 for list of factors.

The core of the statistical analyses is multilevel Bayesian logistic regression with random effects. The analyses appear to be performed correctly.

I have a several concerns about this manuscript.

The authors do not appear to understand some of the literature that they cite. For example, they write (p. 2):

Because human male gametes bearing X or Y chromosomes are equally frequent (being produced by meiosis symmetrically partitioning two sex chromosomes), and because ova bear only X chromosomes, one would expect a sex ratio at conception of exactly ½ [1]

and they cite ([1]) Fisher (1930) for this claim. Fisher’s treatment of the evolution of the sex ratio contains no mention of sex chromosomes, equal segregation, and certainly does not involve a claim of the sex ratio at conception being or expected to be ½. In fact, he wrote (p. 159):

[The attainment of the sex ratio of the equal investment equilibrium via differential mortality of males] is brought about by a somewhat larger inequality in the sex ratio at conception.

There are many articles that could be correctly cited for the claim that one would expect an even sex ratio at conception (see Orzack et al. 2015 for citations in which this claim is made.)

The authors discuss their results and how they are related to the Trivers-Willard hypothesis (TWH) (pp. 11-13). They conclude (p. 12):

One key ramification of the above analysis is that the TWH cannot provide comprehensive account of the range of exogenous factors associated with SRB variation under the kind of circumstances present in our study.

I am skeptical as to the relevance of the TWH to human populations (and those of other species) but the authors’ conclusion is not anchored in the specifics of their results. What are needed are specific analyses of these data that bear on the predictions of the TWH. In this context, the authors mention the study of Zietsch et al. (2020) and claim that:

In particular, Zietsch et al. have demonstrated that there exists neither within-individual SRB auto-correlation (contra Assumption A1) nor similarity in the SRB for children of siblings (contra Assumption A2).

Their study contains no analyses that bear directly on these assumptions as defined by the present authors (p. 11):

Assumption A1. The condition of a mother during parental investment is correlated with the condition of her offspring; in other words, mothers in better conditions have offspring that will be in better conditions.

Assumption A2. The condition of the offspring persists after parental investment ends, 225 and is positively correlated with the offspring’s reproductive success.

In this context, it also appears that the authors have confused the evolutionary past with the evolutionary present. The Zietsch et al. results and those of others do suggest that there is little genetic variation for the sex ratio in human populations. Beyond that, they do not necessarily imply anything about the past influence of the selective process described by the TWH (cf. Orzack and Hardy 2021). The current human sex ratio may reflect the past influence of the TWH dynamic even if that dynamic does not operate currently. That said, while I think that its realized past influence is likely negligible, it is important to note that opinions differ. At minimum, the authors need to do a better job of marshaling evidence for their claim and addressing the claims that the TWH is an important influence of human sex ratios (cf. Navara 2018).

Finally, the authors do not correctly represent some of the prior literature pertaining to environmental influences on the human sex ratio. They write (p. 8)

Using the US dataset, we were able to validate the findings of a number of previous studies regarding the association between the SRB and exogenous factors (Table 3). Specifically, our data suggests that PCBs (polychlorinated biphenyls), aluminium (Al) in air, chromium (Cr) in water and total mercury (Mg) quantity drive the SRB up, while lead (Pb) in soil appears to be associated with a decreased SRB.

This statement implies, for example, that the influence of PCBs on the human sex ratio is resolved. This implication is incorrect for two reasons. The first is that there are conflicting results in the literature, with some showing an increased sex ratio with PCB exposure and others the opposite (Vartiainen et al. 1999; Weisskopf et al. 2003; Mackenzie et al. 2005; Hertz-Picciotto et al. 2008; Terrell et al. 2009, 2011; Nieminen et al. 2013; Leijs et al. 2014). The second reason is that, if anything, a common understanding is that, in fact, PCB exposure is associated with a decrease in sex ratio, not an increase as claimed by the authors. At minimum, the authors need to acknowledge these heterogeneous results and how their results relate to them. Better would be an attempt to explain if and how their results help resolve the discrepancies among studies.

Fisher, R. A. 1930: The Genetical Theory of Natural Selection. Clarendon Press, Oxford.

Hertz-Picciotto, I., T. A. Jusko, E. J. Willman, R. J. Baker, J. A. Keller, S. W. Teplin, and M. J. Charles. 2008: A cohort study of in utero polychlorinated biphenyl (PCB) exposures in relation to secondary sex ratio. Environmental Health 7:1–8.

Leijs, M. M., L. M. van der Linden, J. G. Koppe, K. Olie, W. M. C. van Aalderen, and G. W. ten Tusscher. 2014: The influence of perinatal and current dioxin and PCB exposure on reproductive parameters (sex-ratio, menstrual cycle characteristics, endometriosis, semen quality, and prematurity): a review. Biomonitoring 1:1–15.

Mackenzie, C. A., A. Lockridge, and M. Keith. 2005: Declining sex ratio in a first nation community. Environmental health perspectives 113:1295–1298.

Navara, K. J. 2018: Choosing Sexes : Mechanisms and Adaptive Patterns of Sex Allocation in Vertebrates. Springer International Publishing, Cham, Switzerland.

Nieminen, P., H. Lehtiniemi, A. Huusko, K. Vähäkangas, and A. Rautio. 2013: Polychlorinated biphenyls (PCBs) in relation to secondary sex ratio – A systematic review of published studies. Chemosphere 91:131–138.

Orzack, S. H., and I. C. W. Hardy. 2021: Does the lack of heritability of human sex ratios require a rethink of sex ratio theory? No: a Comment on Zietsch et al. 2020. Proceedings of the Royal Society B 288:20202638.

Orzack, S. H., J. W. Stubblefield, V. R. Akmaev, P. Colls, S. Munné, T. Scholl, D. Steinsaltz, and J. E. Zuckerman. 2015: The human sex ratio from conception to birth. Proceedings of the National Academy of Sciences 112:E2102–E2111.

Terrell, M. L., K. P. Hartnett, and M. Marcus. 2011: Can environmental or occupational hazards alter the sex ratio at birth? A systematic review. Emerging Health Threats 4:7109.

Terrell, M. L., A. K. Berzen, C. M. Small, L. L. Cameron, J. J. Wirth, and M. Marcus. 2009: A cohort study of the association between secondary sex ratio and parental exposure to polybrominated biphenyl (PBB) and polychlorinated biphenyl (PCB). Environmental Health 8:1–12.

Vartiainen, T., L. Kartovaara, and J. Tuomisto. 1999: Environmental chemicals and changes in sex ratio: Analysis over 250 years in Finland. Environmental Health Perspectives 107:813–815.

Weisskopf, M. G., H. A. Anderson1, L. Hanrahan, and The Great Lakes Consortium. 2003: Decreased sex ratio following maternal exposure to polychlorinated biphenyls from contaminated Great Lakes sport-caught fish: a retrospective cohort study. Environmental Health 2:1–14.

Zietsch, B. P., H. Walum, P. Lichtenstein, K. J. H. Verweij, and R. Kuja-Halkola. 2020: No genetic contribution to variation in human offspring sex ratio: a total population study of 4.7 million births. Proceedings of the Royal Society B: Biological Sciences 287:20192849.

**Have the authors made all data and (if applicable) computational code underlying the findings in their manuscript fully available?**

Reviewer #1: **No: **Could you please provide the R code used to analyse this data

Reviewer #2: **No: **I did not see any information about the availability of the raw data. If correct, this information should be provided

PLOS authors have the option to publish the peer review history of their article (what does this mean?). If published, this will include your full peer review and any attached files.

Reviewer #1: No

Reviewer #2: No

Figure Files:

Data Requirements:

Reproducibility:

References:

---

## [Editor Report · Decision Letter 1]

23 Sep 2021

Dear Dr. Rzhetsky,

Thank you very much for submitting your manuscript "Observable Variations in Human Sex Ratio at Birth" for consideration at PLOS Computational Biology. As with all papers reviewed by the journal, your manuscript was reviewed by members of the editorial board and by several independent reviewers. The reviewers appreciated the attention to an important topic. Based on the reviews, we are likely to accept this manuscript for publication, providing that you modify the manuscript according to the review recommendations.

Only minor issues remain, and these do not warrant another round of review.

“In other words, if SRB is ever influenced by some factor(s) at least partially inheritable, then SRB itself would have to be heritable as well which the results from Zietsch et al. rule out.”

>> “Inheritable” has a different meaning from “heritable” – heritable is the appropriate word here.

“Thus, our results are better interpreted as supporting the intrinsic randomness of the SRB and/or the dependence of the SRB on non-adapative (e.g. socio-cultural [71, 77, Ch. 12]), causal factors, possibly including

those common to both changes in the SRB and associated exogenous factors.”

>> It doesn’t quite make sense to say that your results, which involve associations with other variables, support the intrinsic randomness of SRB. If SRB was truly random, then it wouldn’t be associated with other variables. Rephrase, distinguishing 1) the strong influence of random Mendelian randomisation from 2) the SRB itself, which is not quite random, as these results show.

The authors should go through the text carefully checking for grammatical issues, especially the placement of commas. Often there were commas where there shouldn’t be. Also check spelling (e.g. non-adapative in the above quote).

Sincerely,

Brendan Zietsch

Guest Editor

PLOS Computational Biology

Nina Fefferman

Deputy Editor

PLOS Computational Biology

[LINK]

Thank you for these responses and changes the the text. Only minor issues remain, and these do not warrant another round of review.

“In other words, if SRB is ever influenced by some factor(s) at least partially inheritable, then SRB itself would have to be heritable as well which the results from Zietsch et al. rule out.”

>> “Inheritable” has a different meaning from “heritable” – heritable is the appropriate word here.

“Thus, our results are better interpreted as supporting the intrinsic randomness of the SRB and/or the dependence of the SRB on non-adapative (e.g. socio-cultural [71, 77, Ch. 12]), causal factors, possibly including

those common to both changes in the SRB and associated exogenous factors.”

>> It doesn’t quite make sense to say that your results, which involve associations with other variables, support the intrinsic randomness of SRB. If SRB was truly random, then it wouldn’t be associated with other variables. Rephrase, distinguishing 1) the strong influence of random Mendelian randomisation from 2) the SRB itself, which is not quite random, as these results show.

The authors should go through the text carefully checking for grammatical issues, especially the placement of commas. Often there were commas where there shouldn’t be. Also check spelling (e.g. non-adapative in the above quote).

Figure Files:

Data Requirements:

Reproducibility:

References:

---

## [Editor Report · Decision Letter 2]

25 Oct 2021

Dear Dr. Rzhetsky,

We are pleased to inform you that your manuscript 'Observable Variations in Human Sex Ratio at Birth' has been provisionally accepted for publication in PLOS Computational Biology.

Best regards,

Brendan Zietsch

Guest Editor

PLOS Computational Biology

Nina Fefferman

Deputy Editor

PLOS Computational Biology

Thanks to the authors for their attention to the comments. (And yes I meant Mendelian segregation, thank you.) I am pleased to accept the manuscript.

---

## [Editor Report · Acceptance letter]

12 Nov 2021

PCOMPBIOL-D-21-00882R2 

Observable Variations in Human Sex Ratio at Birth

Dear Dr Rzhetsky,

I am pleased to inform you that your manuscript has been formally accepted for publication in PLOS Computational Biology. Your manuscript is now with our production department and you will be notified of the publication date in due course.

With kind regards,

Katalin Szabo
